# Optimized Pretreatment of Non-Thermal Plasma for Advanced Sewage Oxidation

**DOI:** 10.3390/ijerph17207694

**Published:** 2020-10-21

**Authors:** Hee-Jun Kim, Chan-Hee Won, Hyun-Woo Kim

**Affiliations:** 1Soil Environment Research Center, Department of Environmental Engineering, Division of Civil, Environmental, Mineral Resource and Energy Engineering, Jeonbuk National University, 567 Baekje-daero, Deokjin-gu, Jeonju 54896, Korea; rlagml3377@jbnu.ac.kr (H.-J.K.); chwon@jbnu.ac.kr (C.-H.W.); 2Department of Environment and Energy, Jeonbuk National University, 567 Baekje-daero, Deokjin-gu, Jeonju 54896, Korea

**Keywords:** non-thermal plasma, advanced oxidation process, statistical optimization, temperature, contact time, pollutant removal

## Abstract

This study investigates how the non-thermal plasma (NTP) process leads to advanced oxidation of sewage using response surface methodology. For environmentally viable and efficient operation of the NTP process, temperature and contact time were selected as two important independent variables. Their impacts on the performance were tested following an experimental design to figure out optimal operating conditions. Based on obtained treatment efficiency, statistically optimized conditions were derived by using an approach adapting the central composite design. Results show that coupling 40 °C of temperature and 4 h of contact time demonstrate optimal performance for total chemical oxygen demand (TCOD, 59%) and total suspended solids (85%), respectively. This implies that NTP may present efficient particulate destruction leading to organic solids dissolution. Statistical analysis reveals that the contact time shows more significant dependency than the temperature on the advanced oxidation of TCOD, possibly due to dissolved organic material. For total nitrogen removal, on the contrary, the optimal efficiency was strongly related to the higher temperature (~68 °C). This work provides an inroad to considering how NTP can optimally contribute to better oxidation of multiple pollutants.

## 1. Introduction

Applying conventional oxidants such as chlorine or ozone (O_3_) can destroy target pollutants but has a crucial disadvantage in that certain organic substances turn into residual by-products during the oxidation [1,2]. In particular, continuous disinfection with chlorine produces by-products such as trihalomethanes and halo-acetic acids that are problematic for human health [3].To compensate for such drawbacks, advanced oxidation processes (AOPs) have been applied alternatively [4].

Although numerous methods have been attempted such as Fenton oxidation, peroxone, catalyzed UV photolysis, etc., each process has its own specific disadvantages [5] and all still require methods to promote the generation of highly reactive oxidants, for example, hydroxyl (OH) radical. Fenton oxidation, using oxidants formed from ferrous salts plus H_2_O_2_ in acidic solution, produces excessive sludge and demonstrates low efficiency when organic contents are high [6,7]. O_3_ oxidation needs control over pH to reduce the scavenging effect [8]. Peroxone needs proper control over the ratio between O_3_ and hydrogen peroxide (H_2_O_2_) [9]. Catalyzed UV photolysis can be easily inhibited by turbidity or color [10]. Therefore, scale-up for such a process is still a problematic issue despite the great potential for the effective destruction of pollutants [11].

Non-thermal plasma (NTP) is free from these limitations, thus it has received attention recently as a viable alternative to traditional AOPs for treating wastewaters including non-biodegradable, refractory, or even toxic substances [12]. The literature reveals that NTP can effectively lower the ecotoxicity of degradation by-products [13]. A fundamental feature of NTP is that electrical discharges such as dielectric barrier, corona, glow, etc., generate highly reactive chemical species in a carrier gas, air, and they may react with vapor or water molecules, promoting recombination for longer enhanced oxidation [14]. Considerable efforts have been made to apply NTP such as in aerospace engineering, biomedical applications, textile technology, semiconductor manufacturing, and decontamination [15]. Little application, however, has been reported targeting treatment works.

The major reactive oxidants of NTP seem to be either various ions, electrons, radicals, activated molecules, or their long-lasting recombinants (e.g., O_3_ and H_2_O_2_) [16,17,18]. Radicals show rapid reactivity to organic matters, suspended solids, or reduced inorganic pollutants depending on the range of temperature, [19,20] thus the temperature is important for maximum or optimal activity. Previous studies have also shown that longer contact time (CT) and higher temperature are factors that determine the decomposition of organic particulates releasing soluble chemical oxygen demand (SCOD) [21,22]. Since NTP is environmentally-friendly and energy-saving despite stronger oxidation [23], potential exists for the development of a robust treatment process that can handle non-biodegradable, toxic, or abiotic pollutants in wastewater. Despite the promise of NTP, little attention has been paid to the possibility of reducing the organic loading through shorter pretreatment of high concentration wastewater with non-degradable organic material. Little insights have been provided on how NTP contributes to total nitrogen removal according to temperature and CT [24].

To analyze the optimization problem, response surface methodology (RSM) has been frequently used [25] because it suggests an optimal condition of independent variables to obtain the best dependent response to the problem, using a useful collection of mathematical and statistical techniques [26]. Among various experimental designs, central composite design (CCD) is useful in building a second-order response surface model with two independent variables and with a rotatable experimental design working from a center [25]. Previous studies have shown that CT and temperature are both important variables in NTP application [21,22] but little information is available regarding simultaneous optimization using statistical and mathematical approaches.

The purpose of this study, therefore, is to provide a framework for figuring out how NTP results in the advanced oxidation of pollutants in wastewater. To determine the optimal pretreatment condition of NTP application, this study adapts response surface methodology using CCD. The effects of the temperature and the CT on the treatment performance were investigated based on degradation kinetics. In addition, this study illustrates which mechanisms are associated with the responses to NTP operation.

## 2. Materials and Methods

### 2.1. Characteristics of Wastewater

Table 1 presents the physical, chemical, and biological characteristics of the wastewater used in this study. The wastewater was sampled from a flow adjustment tank in a publicly owned treatment work (POTW) at N-city, Korea.

### 2.2. Experimental Set-up of the NTP

A lab-scale of glow-discharge NTP system (Groon Co., Ltd., Jeonju, Korea) was manually manufactured for the experiments and had an electrical specification of 10 mA and 2.2 W. An aeration pump (Goseong valve, YP−20A, Korea) was connected to the NTP system to supply the air at room temperature (about 25 °C) with a maximum air flowrate of 20 L/min. The air flowrate was controlled by a flow meter (Dwyer, RMA−22-SSV, Michigan City, IN, USA). Two rubber lines were made on the NTP system, and the airflow was set at 10 L/min by inserting a round sparger (diameter 3 cm) into the reactor. The wastewater was reacted in a 2.5 L bottle (2 L working volume). For homogeneous radical contact in the wastewater, the reactor was vigorously mixed by using an agitator (DAIHAN Scientific, MSH−20A, Wonju, Korea). The temperature was adjusted according to an experimental design by using a water bath (DAIHAN Scientific, LSB−055S, Wonju, Korea). Figure 1 shows a schematic of the bench-scale NTP system.

### 2.3. Operating Procedures of NTP

NTP was operated to conduct batch experiments. To test the temperature effect, a water bath equipped with a thermometer (DAIHAN Scientific, thermometer with mercury 0~100 °C, Korea) was used. Samples for physical, chemical, and biological analyses were inserted in a 50 mL vial following the sampling schedule at 0 h, 1.2 h, 2 h, 4 h, 6 h, or 6.8 h. The temperature of the reactor was set to 40 °C as a center point, and varied according to the experimental design (12~68 °C). CT was similarly differentiated between 1.2 h and 6.8 h.

### 2.4. Experimental Design

To verify the optimal operating condition of NTP, a CCD was applied by setting the temperature (°C) of the reactor and the CT (hr) as two independent variables, which have been known to control degradation kinetics. Their effects on the pollutant removal rates were set as dependent variables, which constitute a quadratic response surface for each dependent variable. The DOE based on actual variations of independent variables and coded variables, normalized to 1 or −1 as shown in Table 2 and Figure 2, optimizes the responses from the combination of two groups of design points according to multiple response algorithms in response surface methodology [27]. Since the temperature range is determined according to the set levels of a coded variable, the center point was set at 40 °C to prevent the temperature from being set to below zero or too high. For CT, the center point was set at 4 h to guarantee the effectiveness of pretreatment. All the experiments were conducted in triplicate. Computer software, Design-Expert, was used for the statistical assessments of the experimental design [28,29].

Based on the monitoring data of total chemical oxygen demand (TCOD), total suspended solids (TSS), total nitrogen (TN), and pH according to the sampling plan, regressions were conducted to estimate reaction rates of the degradation kinetics by assuming the first-order reaction. Obtained degradation rates were used as the dependent responses for the statistical optimization in order to promote the decontamination.

### 2.5. Statistical Analysis and Regressions

Analyses of variance (ANOVA) were carried out using the Design-Expert software (version 7.0.0; Stat-Ease, Inc., Minneapolis, MN, USA). ANOVA tests variability between experiments and whether there are differences in the experimental means using F statistic and *p*-value. SigmaPlot 12.0 (Systat Software, Inc., San Jose, CA, USA) was used for the optimization studies. In the exponential decay, the decay rate constant (k) was regressed using the equation y = y_0_ + ae^(−k x)^ where y is concentration (mg/L), x is CT (h), and the obtained k values for COD, TSS, and TN were used as dependent variables.

### 2.6. Analytical Methods

Adapting standard methods for the examination of water and wastewater [30], we determined TCOD concentration with method number 5220 B, TN concentration with method number 4500-N, and TSS concentration with the method number 2540 D. pH was analyzed using a pH meter (LAQUA F−74, HORIBA, Japan).

## 3. Results and Discussion

### 3.1. Features of NTP During Wastewater Treatment

Using the NTP system, this study evaluates NTP’s impacts on effluent quality by variation of temperature and CT. The main hypothesis is that the dynamics of pH, TOC, TSS, and TN in the effluent are strongly interrelated with temperature and CT for optimal performance. Thus their changes were monitored as important parameters.

#### 3.1.1. pH

Figure 3 shows the change in pH during the NTP application. For all the experiments, inclined pH was monitored. ∆pH positively increased by about 1.5 to 2 regardless of operating conditions. Generated OH radicals from NTP must have reacted instantaneously with electron donors and contributed to the increase in pH. It was observed that the higher temperature condition lead to a more inclined ∆pH. Run 7 at 68 °C showed a maximum value of pH 9.6 and run 8 at 60 °C showed a value of pH 9.5 while run 12 at 14 °C showed a value of pH 8.8, and run 3 at 20 °C showed a value of pH 8.4.

In general, when the temperature is high, dissociation of water slightly increases, thus leading to a decreased pH. However, on the contrary, increased pH in this study supports the active production of OH radical while the produced H^+^ is buffered by the carbonate system originating from aeration and pollutant oxidation [31]. By the neutralization action, equilibrium must have been induced, keeping pH constant.

#### 3.1.2. TCOD and TSS

Figure 4 shows the change in TCOD and TSS during the NTP treatment of the wastewater. In the case of the CT variation experiment, we found that the concentration of TCOD decreased as the CT became longer. Run 1 (40 °C) presented a gradual TCOD decrease of 4 h. Similar trends were observed at Run 3 (20 °C) and run 8 (60 °C). It was found also that too high temperature significantly lowers the TCOD removal efficiency. The lowest temperature of run 12 (12 °C, 4 h) obtained 58.2%, while run 7 (68 °C, 4 h), the highest temperature condition, obtained only 3%, which was the lowest removal efficiency. The difference between run 7 and run 12 was as much as 59%. These results indicate that adequate temperature control below the central point (40 °C, 4 h) is helpful for NTP to obtain better removal efficiency.

The maximum removal efficiency of TSS reached 84.5% at the center point (40 °C and 4 h) and run 9 (20 °C and 2 h) showed the lowest value (14.6%). CT difference (1 h and 7 h) demonstrates a dramatic difference in TSS removal efficiencies of 37.8% and 75.7%, even at the same temperature (40 °C), respectively. It seems that long CT is preferable for better TSS treatability, but the same CT (4 h) with a range of temperature shows better performance, since different temperatures (12 °C, 40 °C, and 68 °C) recorded removal efficiency as 84.5%, 84.5%, and 20.4%, respectively.

This result corresponds to the earlier studies indicating that the reactive chemicals such as radicals may have higher activity at a relatively lower temperature (<45 °C) than at a higher temperature which leads to significant differences in degradation rates [19]. Run 12 and all the center runs presented increasing or maintaining trends of TCOD, possibly due to SCOD release from the destroyed solids [22]. The longer the CT, the greater the TCOD removal efficiency by NTP [14]. For solids, the hydrolysis step is a rate-limiting step, which was consistent with our experimental result. TCOD and TSS removal must have been associated with the strong oxidizing power of reactive chemicals [32,33,34]. These results indicate that proper adjustment of the NTP’s operating temperature and CT at around 40 °C and longer than 4 h may lead to better TCOD and TSS removal.

#### 3.1.3. TN

Figure 5 shows the dynamics of TN during and after NTP treatment. Comparing the removal efficiencies of different temperatures, run 12, the center runs, and run 7 at the same CT (4 h) presented 9.5%, 30.9%, and 68.5%, respectively. Run 3 (20 °C, 6 h) and run 8 (60 °C, 6 h) also indicated starkly different removal efficiencies (7.5% and 78.6%). In contrast to TCOD and TSS, a high-temperature condition demonstrates better TN removal efficiency. Run 8 (60 °C) and run 6 (68 °C) showed 74.3% and 68.9%, respectively, at the same 4 h CT, which means the high temperature is suitable for TN removal, possibly due to the decreased solubility of ammonia.

Previously, it was indicated that nitrogen removal may be accelerated as the temperature increases [35,36]. Other researchers [37,38] also reported similar results at high pH (pH range 10.5–11) and high temperature (40–80 °C) [39]. Although ammonia gas stripping by controlling pH and temperature is the most widely used AOP for nitrogen removal from wastewater, when reactive chemical species from NTP are abundant, another possible mechanism for TN removal may be partial selective catalytic oxidation/reduction reactions (SCR/SCO) or wet oxidation [23]. However, various forms of nitrogen species make it hard to trace how these mechanisms convert organic/inorganic nitrogen into nitrogen gas [23,40].

### 3.2. Optimization of Pollutant Removal by Central Composition Design

To investigate optimal temperature and CT conditions for the NTP system of treating wastewater, experimental responses of CCD were obtained by statistical DOE software. Based on the hypothesis that a high removal rate easily leads to a high removal efficiency, the rate constants were estimated by regression on experimentally determined data.

Table 3 shows the regressed values of removal rate constant (k) for each experiment. Estimated k values were significantly different from each other. In the case of TCOD, k ranged between 0.0073 h^−1^ and 1.4181 h^−1^. The k values of TSS and TN ranged from 0.0575 h^−1^ to 1.3817 h^−1^, and from 0.0256 h^−1^ to 1.0838 h^−1^, respectively. These large differences support the necessity of optimization of the operating condition.

Using the data from Table 3, empirical polynomial models were constructed after ANOVA. The importance of the model can also evaluate by considering either its F-values or the *p*-values. The F-values are decided by the ratio of the mean square of the parameter during research to the mean square of the error term, while the *p*-values can be calculated from the F distribution.

#### 3.2.1. Optimal Operating Condition for TCOD Removal

Figure 6 a presents a response surface of TCOD removal rate constants according to the combination of temperature and CT. The surface plot indicates that the maximum removal rate can be obtained at around the central point (40 °C and 4 h). In addition, Table 4 shows ANOVA test results based on the experimental results. Degrees of freedom (Df) represents the total amount of data analyzed. As considered in the ANOVA theory, it is clear that the larger value of F acts as a key factor that proves the effect of independent variables [41]. Thus, the model F-value of 34.04 implies that the model is highly significant. There is only a 0.05% chance that the large F-value could occur due to noise. Also, the terms temperature^2^, CT, and CT^2^ were found to be significant model terms. The *p*-value of the suggested quadratic model was lower than 0.05, which supports the statistical significance of the model.

The *p*-value of temperature was higher than 0.05 while that of temperature^2^ was less than 0.05, which means that the relationship between the temperature and TCOD removal is complicated rather than linear. For CT, both *p*-values of CT and CT^2^ term presented a statistically significant relationship and the lower *p*-value of CT^2^ indicates its complicated relationship with TCOD removal. With the result of ANOVA, the following model (1) was derived to estimate the TCOD removal rate with high confidence.
TCOD = 1.26 − 0.03 temperature − 0.12 CT + 0.11 temperature CT − 0.60 temperature^2^ − 0.32 CT^2^(1)

#### 3.2.2. Optimal Operating Condition for TSS Removal

Figure 6b presents a response surface using a combination of independent variables to maximize the TSS removal rate of NTP. It seems that the central points are close to the optimal combination. Table 5 presents the model F-value of 3.12 implying only an 8.48% chance that the model F-value could occur due to noise. In Table 6, the *p*-value of the TSS model was also greater than 0.05, so the unexplainable noise factor of temperature and CT is significant. However, the terms temperature^2^ and CT^2^ were found to be the most significant model terms to explain the TSS removal rate. Statistically, it can be confirmed that the *p*-value of the temperature^2^ and CT^2^ is less than 0.05, which shows their complicated correlation. Equation (2) shows the derived model equation:TSS = 1.32 − 0.054 Temperature − 0.048 CT + 0.045 Temperature CT − 0.45 Temperature^2^ − 0.39 CT^2^(2)

#### 3.2.3. Optimal Operating Condition for TN Removal

Figure 6c illustrates a response surface based on a combination of independent variables to verify NTP’s effect on the TN removal rate. It can be seen that run 7 is the most optimal combination within the ranges of CT and Temperature tested in this study. Table 6 shows the ANOVA test results based on the experimental results indicating that the model is accurate, and that temperature is the only significant model term. The *p*-value of the suggested quadratic model was lower than 0.05, which supports the statistical significance of the model. Temperature was revealed as the most significant factor for TN removal since other terms’ *p*-values exceeded 0.05, which indicates little statistical correlation. For TN removal, therefore, only the temperature has a linear influence. The suggested model Equation (3) can reasonably predict the removal rate of TN.
TN = 0.27 + 0.29 Temperature + 0.023 CT + 0.052 Temperature CT + 0.10 Temperature^2^ − 0.085 CT^2^(3)

#### 3.2.4. Overlay Plot for Process Optimization

Using the three model Equations (1)–(3) from the previous section, we illustrated an overlay plot to consider the three responses together and to provide a better perception of the optimized operating conditions graphically (Figure 6d). By overlaying these response surfaces, a contour plot was generated to verify optimal conditions to maximize TCOD, TSS, and TN removal rates simultaneously. We recognize a common optimum region for independent variables, temperature and CT, in this study. The yellow area in Figure 6d demonstrates that the region fits the proposed optimization criteria corresponding to removal rates of TCOD, TSS, and TN which are k > 0.5 h^−1^ simultaneously. The minimum temperature is about 37.2 °C, with a contact time of about 1 hr. The revealed optimal condition was found to be 55.2–58.2 °C for temperature and 1 h 43 min–2 h 43 min for CT.

Table 7 compares NTP’s performance with other AOPs to confirm the superiority of NTP for wastewater treatment. It was confirmed that, overall, NTP’s removal efficiencies for TCOD, TSS, and TN show about 20.7–43.7% higher when compared to the literature. This optimization approach thus provides a way to economically control problematic sewage in a POTW suffering from non- or slowly biodegradable pollutants.

## 4. Conclusions

This study verified the optimal operating condition of the NTP process for better removal rate, constant with the assumption that ta high degradation rate would lead to a high degradation efficiency. TCOD and TSS showed optimum efficiency at 40 °C and 4 h CT while TN showed optimum efficiency at 68 °C regardless of CT. TN removal was effective at high temperatures (>40 °C). Results demonstrate that the optimal temperature and CT can be varied depending on the wastewater pollutants, thus a proper optimization strategy is highly necessary to figure out the best performance when applying NTP. This study suggests the operation guidelines of NTP to reduce pollutant loadings in wastewater and also provides the framework for suitable process optimization for target pollutant removal using the statistical DOE approach.

## Figures and Tables

**Figure 1 ijerph-17-07694-f001:**
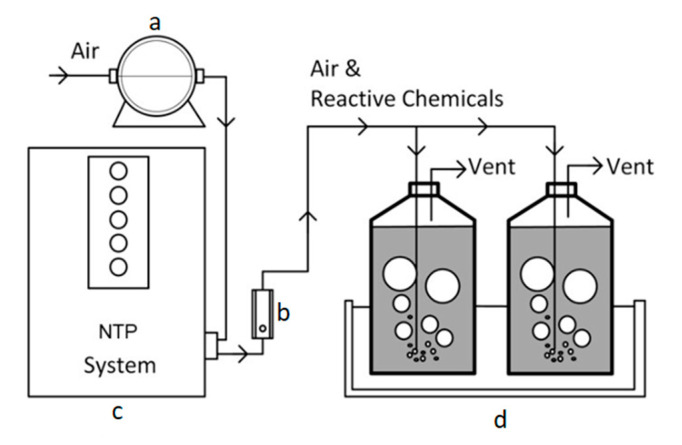
Lab-scale set-up of NTP system: (**a**) aerator, (**b**) flow meter, (**c**) glow discharge NTP generator, and (**d**) bottle and water bath.

**Figure 2 ijerph-17-07694-f002:**
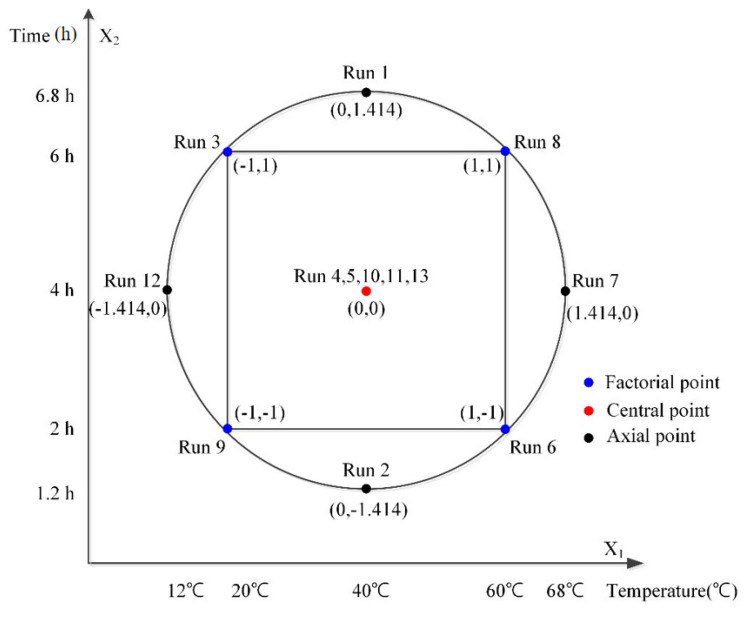
Schematic diagram of central composite design (CCD) as a function of X_1_: Temperature (°C) and X_2_: Time (h) according to the four factorial point, five central points, and four axial points. (Fixed factor: reactor volue = 2 L, flow rate 10 L/min).

**Figure 3 ijerph-17-07694-f003:**
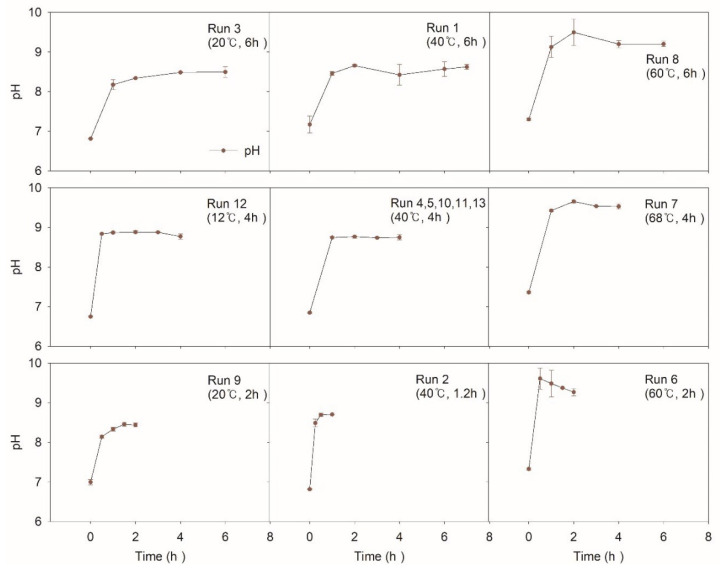
pH value according to experimental conditions.

**Figure 4 ijerph-17-07694-f004:**
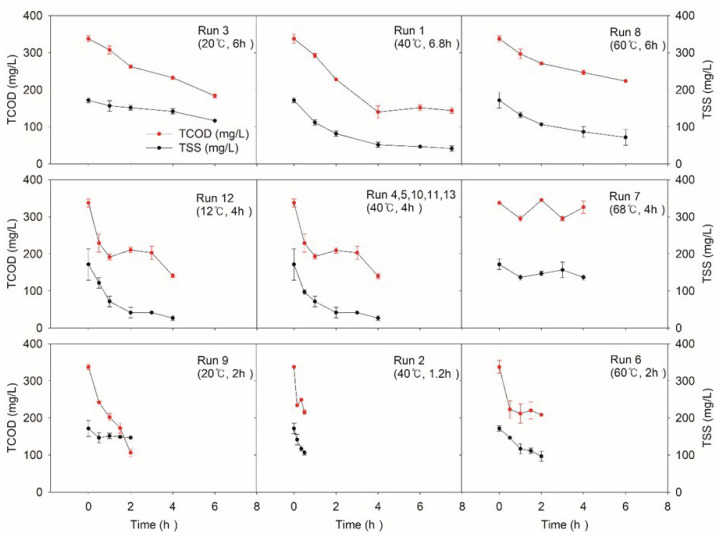
Variations of total chemical oxygen demand (TCOD) and total suspended solids (TSS) according to the experimental condition of central composite design (CCD). The average values of initial TCOD and TSS were 337. 5 mg/L and 171.6 mg/L, respectively.

**Figure 5 ijerph-17-07694-f005:**
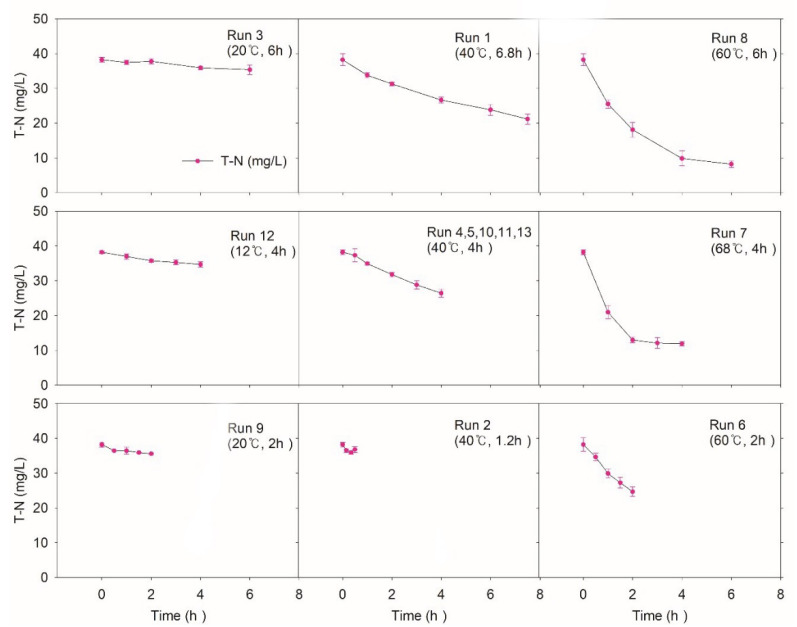
Variations of total nitrogen (TN) according to the experimental condition of CCD. The initial values of TN were 38.2 mg/L.

**Figure 6 ijerph-17-07694-f006:**
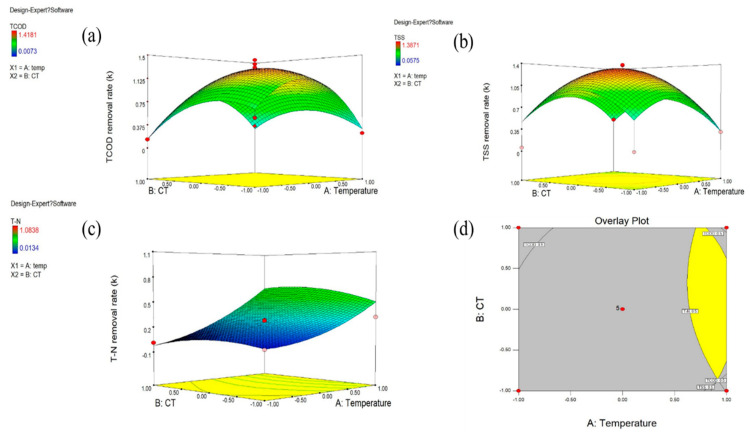
Optimal Simulation by CCD Experiment plan and overlay plot: (**a**) TCOD removal rate. (**b**) TSS removal rate. (**c**) TN removal rate. (**d**) The optimal condition for each experiment.

**Table 1 ijerph-17-07694-t001:** Characteristics of substrate sampled from a flow adjustment tank in a POTW.

Item	Unit	Average	Standard Deviation
Total chemical oxygen demand (TCOD)	mg/L	337.5	61.3
Total nitrogen (TN)	mg/L	38.2	1.9
Total phosphorus (TP)	mg/L	8.5	1.7
Total suspended solids (TSS)	mg/L	171.0	50.5
pH	-	6.8	0.1
Conductivity	S/m	8.7 × 10^3^	0.3 × 10^3^
Total Coliforms	cells/mL	135,000	-

**Table 2 ijerph-17-07694-t002:** Independent variables and their levels of experimental design.

Independent Variables	Symbol	Unit	Levels of Coded ^a^ and Actual Variable				
			−1.414	−1	0	1	1.414
Temperature	X_1_	°C	12	20	40	60	68
CT	X_2_	h	1.2	2	4	6	6.8

^a^ Coded variable of temperature (X_1_) and contact time (X_2_) were calculated based on actual levels of variables by the following equations: C_1_ = (X_1_ − 40)/20 and C_2_ = (X_2_ − 40)/20.

**Table 3 ijerph-17-07694-t003:** The removal rate k and coefficient of determination (R^2^) regressed using the data obtained from each experimental condition for CCD.

Run	Removal Rate, k (h^−1^)			Coefficient of Determination, R^2^		
	TCOD	TSS	TN	TCOD	TSS	TN
1	0.40	0.60	0.18	0.96	0.99	0.99
2	0.90	0.99	0.17	0.72	0.98	0.88
3	0.14	0.06	0.01	0.99	0.96	0.92
4	1.42	1.39	0.26	0.87	0.99	0.99
5	1.10	1.38	0.28	0.86	0.97	0.99
6	0.24	0.30	0.32	0.62	0.97	0.99
7	0.01	0.30	1.08	0.02	0.29	0.99
8	0.36	0.47	0.51	0.99	0.99	0.99
9	0.47	0.06	0.03	0.97	0.52	0.81
10	1.12	1.18	0.28	0.85	0.97	0.98
11	1.35	1.29	0.28	0.96	0.97	0.99
12	0.17	1.07	0.03	0.67	0.99	0.96
13	1.30	1.39	0.27	0.96	0.99	0.98

**Table 4 ijerph-17-07694-t004:** ANOVA test result for response surface quadratic model for TCOD removal by CCD.

Source	Sum of Squares	Df	Mean Square	F Value	*p*-Value Prob > F
Model	3.09	5	0.62	34.04	<0.0001
Temperature	0.007	1	0.007	0.41	0.5447
CT	0.11	1	0.11	5.86	0.0460
Temperature × CT	0.050	1	0.050	2.76	0.1406
Temperature^2^	2.52	1	2.52	138.51	<0.0001
CT^2^	0.71	1	0.71	39.13	0.0004

**Table 5 ijerph-17-07694-t005:** ANOVA test result for response surface quadratic model for TSS removal by CCD.

Source	Sum of Squares	Df	Mean Square	F Value	*p*-Value Prob > F
Model	2.24	5	0.45	3.12	0.0848
Temperature	0.023	1	0.023	0.16	0.7001
CT	0.018	1	0.018	0.13	0.7331
Temperature × CT	0.008	1	0.008	0.057	0.8181
Temperature^2^	1.40	1	1.40	9.76	0.0167
CT^2^	1.08	1	1.08	7.49	0.0291

**Table 6 ijerph-17-07694-t006:** ANOVA test result for response surface quadratic model for TN removal by CCD.

Source	Sum of Squares	Df	Mean Square	F Value	*p*-Value Prob > F
Model	0.81	5	0.16	10.53	0.0037
Temperature	0.65	1	0.65	42.31	0.0003
CT	0.004	1	0.004	0.28	0.6162
Temperature × CT	0.011	1	0.011	0.71	0.4284
Temperature^2^	0.076	1	0.076	4.90	0.0625
CT^2^	0.050	1	0.050	3.27	0.1136

**Table 7 ijerph-17-07694-t007:** Comparison of treatment efficiency in TCOD, TSS, and TN between non-thermal plasma (NTP) and other advanced oxidation processes (AOPs).

Division	Temperature (°C)	Contact Time (h)	Treatment Type	Wastewater Source	Removal Efficiency (%)	Reference
TCOD	30	3	O_3_	Landfill leachate	14.5	[42]
			O_3_/H_2_O_2_		35.0	
	40	6	NTP	Sewage	58.2	This study
TSS	-	4	Ti/β-PbO_2_	Textile	63.8	[43]
	-	2	Photo-Fenton-Electrocoagulation	Tannery effluent	65.0	[44]
	40	6	NTP	Sewage	84.5	This study
TN	22−25	6	Ti/Pt/PbO_2_	Landfill leachate	~40	[45]
			Ti/Pt/SnO_2_-Sb_2_O_4_		~35	
	60		NTP	Sewage	78.6	This study

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
