# Peer review of "Optimized Pretreatment of Non-Thermal Plasma for Advanced Sewage Oxidation"

_ijerph, 2020, doi:10.3390/ijerph17207694_

Round 1
Reviewer 1 Report
In this manuscript, non-thermal plasma (NTP) process was investigated for advanced oxidation of sewage using response surface methodology. Temperature and contact time were selected as two important independent variables that were optimised using central composite design. Results of this study are interesting but following comments need to be addressed before the acceptance of this manuscript.
Line 31-34: AOP by NTP would also produce transformation by-products. It is important to highlight the toxicity of by-products, particularly following chlorination.
Line 52: H2O2 is not a radical – please revise.
This study employs response surface methodology (central composite design). A brief overview of this methodology and why central composite design was selected should be made part of the introduction section. In addition, please focus on the use of RSM based on the previous studies and how it can help optimize operating conditions.
Table 1: please define COD(cr) in the caption. In addition, consider writing TP and TN instead of T-N and T-P here and elsewhere in the manuscript. Define the abbreviation at their first appearance.
Feed pH is an important operating parameter for AOPs – It would had been interesting to include this variable in the experimental design i.e., Box Behnken. Please provide suitable explanation in the manuscript.
Since the duration for each experiment was not long, did the author replicate their experiment for reproducibility?
Results and discussion section lacks appropriate justifications of the obtained results, and have not been explained considering previous studies. If the relevant studies are not available, the authors are suggested to compare the results with other AOPs.
Please discuss the significance of different parameters (ANOVA analysis) such as Df and F value etc.
Please consider improving the visibility of Figure 6.
Please add a brief paragraph on the practical implementation of the obtained results at full-scale, particularly process economy
Please cite at least 4-5 new references from last two years.
Author Response
POINT-BY-POINT RESPONSE TO EACH REVIEWER’S COMMENTS
Authors: Hee-Jun Kim, Chan-Hee Won, and Hyun-Woo Kim
Manuscript ID: ijerph-920819
Manuscript Title: Optimized pretreatment of non-thermal plasma for advanced sewage oxidation
We appreciate the many helpful comments from the reviewer and revised the manuscript following the reviewer’s comments as we indicate below. Except for trivial changes, we highlight our revisions using red color in the manuscript and in this response memo, as suggested by the editor.
Reviewer #1:
In this manuscript, non-thermal plasma (NTP) process was investigated for advanced oxidation of sewage using response surface methodology. Temperature and contact time were selected as two important independent variables that were optimised using central composite design. Results of this study are interesting but following comments need to be addressed before the acceptance of this manuscript.
Line 31-34: AOP by NTP would also produce transformation by-products. It is important to highlight the toxicity of by-products, particularly following chlorination.
Response: We address Reviewer 1’s concern by adding a sentence to show the negative effect of chlorination as shown in the Introduction section of Line 33. And we added a sentence to highlight the positive aspect of NTP because our previous work revealed that NTP can lower down the ecotoxicity of by-products eventually in the end.
To better communicate these points, we revised the below sentences:
(Page 1 Line 33) “Applying conventional oxidants such as chlorine or ozone (O3) can destroy target pollutants but it has a crucial disadvantage that certain organic substances turn into residual by-products during the oxidation [1,2]. In particular, continuous disinfection with chlorine produces chlorine disinfection by-products such as trihalomethanes and haloacetic acids that it is problematic for human health [3]. To compensate for such drawbacks, advanced oxidation processes (AOPs) have been applied alternatively [4].”
(Page 2 Line 48) “Literature revealed that NTP can effectively lower down the ecotoxicity of degradation by-products [13].”
(Reference) 13. Lee, D.; Lee, J.-C.; Nam, J.-Y.; Kim, H.-W. Degradation of sulfonamide antibiotics and their intermediates toxicity in an aeration-assisted non-thermal plasma while treating strong wastewater. Chemosphere 2018, 209, 901-907, doi:https://doi.org/10.1016/j.chemosphere.2018.06.125.
Line 52: H2O2 is not a radical – please revise.
Response: The reviewer’s point is well taken. We now revise the expression and add relevant references as shown below to be clearer about the reactive chemical species:
(Page 2 Line 55) “The major reactive oxidants of NTP seems to be either various ions, electrons, radicals, activated molecules, or long-lasting their recombinants (e.g. O3 and H2O2) [16-18]”
(References) “16. Jiang, B.; Zheng, J.; Qiu, S.; Wu, M.; Zhang, Q.; Yan, Z.; Xue, Q. Review on electrical discharge plasma technology for wastewater remediation. Chemical Engineering Journal 2014, 236, 348-368, doi:https://doi.org/10.1016/j.cej.2013.09.090.
- Sato, M.; Ohgiyama, T.; Clements, J.S. Formation of chemical species and their effects on microorganisms using a pulsed high-voltage discharge in water. IEEE Transactions on Industry Applications 1996, 32, 106-112, doi:10.1109/28.485820.
18.Yadav, D.K.; Adhikari, M.; Kumar, S.; Ghimire, B.; Han, I.; Kim, M.-H.; Choi, E.-H. Cold atmospheric plasma generated reactive species aided inhibitory effects on human melanoma cells: an in vitro and in silico study. Scientific Reports 2020, 10, 3396, doi:10.1038/s41598-020-60356-0.”
This study employs response surface methodology (central composite design). A brief overview of this methodology and why central composite design was selected should be made part of the introduction section. In addition, please focus on the use of RSM based on the previous studies and how it can help optimize operating conditions.
Response: Following the reviewer’s opinion, we reinforced the Introduction section by adding a paragraph briefly explaining RSM and CCD.
(Page 2, Line 68) “To analyze the optimization problem, response surface methodology (RSM) has been frequently used [25] because it suggests an optimal condition of independent variables to obtain optimal dependent response of the problem using a useful collection of mathematical and statistical techniques [26]. Among various experimental design, central composite design (CCD) is useful in building second-order response surface model with two independent variables with a rotatable experimental design from a center [25]. Previous studies have shown that CT and temperature are both important variable in NTP application [21,22] but little information is available regarding the simultaneous optimization a using statistical and mathematical approaches”
Table 1: please define COD(cr) in the caption. In addition, consider writing TP and TN instead of T-N and T-P here and elsewhere in the manuscript. Define the abbreviation at their first appearance.
Response: According to the reviewer's opinion, we defined the acronyms including TCOD in Table 1. Also, as suggested by the reviewer, we revised all the T-P and T-N to TP and TN, respectively, as shown below.
(Table 1)
Table 1. Characteristics of substrate sampled from a flow adjustment tank in a POTW.
|
Item |
Unit |
Average |
Standard deviation |
|
Total chemical oxygen demand (TCOD) |
mg/L |
337.5 |
61.3 |
|
Total nitrogen (TN) |
mg/L |
38.2 |
1.9 |
|
Total phosphorus (TP) |
mg/L |
8.5 |
1.7 |
|
Total suspended solids (TSS) |
mg/L |
171.0 |
50.5 |
|
pH |
- |
6.8 |
0.1 |
|
Conductivity |
S/m |
8.7´103 |
0.3´103 |
|
Total Coliforms |
cells / mL |
135,000 |
- |
(Line 123, 140, 143, 149, 197, 198, 202, 203, 213, 222, 265, 267, 272, 273, 274, 275, 277, 280, 285, 288, 294, 295, 299, 300) T-N à “TN”
Feed pH is an important operating parameter for AOPs – It would had been interesting to include this variable in the experimental design i.e., Box Behnken. Please provide suitable explanation in the manuscript.
Response: Agreeing that pH is a topic of great interest and we are aware of its importance in general. However, the pH value of the sewage that we tested is relatively stable in most cases thus we neglected it from the independent variables for optimization. Moreover, using three variables at the same time and controlling pH throughout the experiment is technically difficult though we admit the effectiveness of the Box-Behnken design. We monitored the natural change of pH by NTP application by considering pH as a dependent variable, and discussed the observed phenomenon since we do not want to divert the attention of the reader to the artificial pH impact.
In order to better explain this point of view, we strengthen the sentences as shown below:
(Page 7, Line 204) “Other researchers [38,39] also reported similar results at high pH (pH range 10.5 ~ 11) and high temperature (40~80 ℃) [40]. Although the ammonia gas stripping by controlling pH and temperature is the most widely used AOPs for nitrogen removal from wastewater, when reactive chemical species from NTP are abundant, another possible mechanism for TN removal…”
Since the duration for each experiment was not long, did the author replicate their experiment for reproducibility?
Response: Yes, it is clear from the figures presenting average values with standard deviations that we conducted a triplicate experiments for all the experiments for the accuracy of data. To supply the information, we revised the text as below:
(Page 3, Line 119) “All the experiments were conducted in triplicate.”
The results and discussion section lacks appropriate justifications of the obtained results, and have not been explained considering previous studies. If the relevant studies are not available, the authors are suggested to compare the results with other AOPs.
Response: It appears that we have the same point of view as the reviewer on the justification issue. Since it was very hard for us to find out similar experimental set-up, we compared the overall treatment efficiency of NTP with those of other AOPs in Table 7 to support the excellency of NTP. It seems that we already addressed the reviewer’s concern in the Table.
Please discuss the significance of different parameters (ANOVA analysis) such as Df and F value etc.
Response: We have reinforced the section of 3.2.1. according to the opinion of the reviewer.
(Page 4, Line 137) “ANOVA tests the variability between experiments whether there are differences in the experimental means or not using F statistic and p-value.”
(Page 4, Line 237) “In addition, Table 4 shows ANOVA test results based on the experimental results. The meaning that degrees of freedom (Df) represents the total amount of data analyzed. As considered in the ANOVA theory, it is clear that the larger value of F acts as a key factor that proves the effect of independent variables [42]. Thus, the model F-value of 34.04 implies the model is highly significant.”
Please consider improving the visibility of Figure 6.
Response: The resolution of Figure 6 was improved as requested by the reviewer.
Please add a brief paragraph on the practical implementation of the obtained results at full-scale, particularly process economy
Response: Due to the limitations on manuscript length, we decided to insert an implication sentence as shown below:
(Page 11, Line 296) “This optimization approach thus provides a way to economically control problematic sewage of POTW suffering from non- or slowly- biodegradable pollutants.”
Please cite at least 4-5 new references from last two years.
Response: As suggested by the reviewer, we cited additional references as shown below:
(References)
“3. Mazhar, M.A.; Khan, N.A.; Ahmed, S.; Khan, A.H.; Hussain, A.; Changani, F.; Yousefi, M.; Ahmadi, S.; Vambol, V. Chlorination disinfection by-products in Municipal drinking water–A review. Journal of Cleaner Production 2020, 123159.
- Lee, D.; Lee, J.-C.; Nam, J.-Y.; Kim, H.-W. Degradation of sulfonamide antibiotics and their intermediates toxicity in an aeration-assisted non-thermal plasma while treating strong wastewater. Chemosphere 2018, 209, 901-907, doi:https://doi.org/10.1016/j.chemosphere.2018.06.125.
- Yadav, D.K.; Adhikari, M.; Kumar, S.; Ghimire, B.; Han, I.; Kim, M.-H.; Choi, E.-H. Cold atmospheric plasma generated reactive species aided inhibitory effects on human melanoma cells: an in vitro and in silico study. Scientific Reports 2020, 10, 3396, doi:10.1038/s41598-020-60356-0.
- Cao, L.; Wang, J.; Xiang, S.; Huang, Z.; Ruan, R.; Liu, Y. Nutrient removal from digested swine wastewater by combining ammonia stripping with struvite precipitation. Environmental Science and Pollution Research 2019, 26, 6725-6734.
- Mansouriieh, N.; Sohrabi, M.R.; Khosravi, M. Optimization of profenofos organophosphorus pesticide degradation by zero-valent bimetallic nanoparticles using response surface methodology. Arabian Journal of Chemistry 2019, 12, 2524-2532.”
Reviewer #2:
This study optimized the non-thermal plasma condition on pollutant oxidation. The effects of temperature and contact time were examined by using the response surface method. The experimental design is appropriate and the results is valuable. However, there were still some mistakes in the manuscript and the related mechanism (e.g. the analysis of the radical) is missed. The author should give related discussion in detail when considering resubmission of their work.
The reviewer’s point on context is well taken. Since we already gave full responses to
Specific comments:
1.The concept of non-thermal plasma should be explained in the Introduction section.
Response: The reviewer’s point on context is well taken. Since we already gave information regarding the concept of non-thermal plasma, we do not repeat all of it again here. In brief, our main foci are on the systemic and statistical evaluation of optimal NTP operating condition. Details are explained from line 46 to line 53 in Introduction section. To be clearer about the concept, we revised our explanations in Introduction as shown below.
(Page 2, Line 46) “Non-thermal plasma (NTP) is free from those limitations thus it has received attention recently as a viable alternative to traditional AOPs treating wastewaters including non-biodegradable, refractory, or even toxic substances [12]. The literature revealed that NTP can effectively lower the ecotoxicity of degradation by-products [13]. A fundamental feature of the NTP is that electrical discharges such as dielectric barrier, corona, glow, etc. generate highly reactive chemical species in a carrier gas, air, and they may react with vapor or water molecules promoting recombination for longer enhanced oxidation [14]. Considerable efforts have been made to apply this NTP such as aerospace engineering, biomedical, textile technology, semiconductor manufacturing, and decontamination [15]. Little application, however, has been reported targeting for treatment works.
The major reactive oxidants of NTP seems to be either various ions, electrons, radicals, activated molecules, or long-lasting their recombinants (e.g. O3 and H2O2) [16-18].”
2.Line 86 Figure 1. “Reactive chemicals” was noted in figure 1, but there was no explanation about which kind of chemicals was added.
Response: The reviewer’s point is well taken. We now revise the expression and add relevant references as shown below to be clearer about the reactive chemical species indicated in Figure1:
(Page 2 Line 55) “The major reactive oxidants of NTP seems to be either various ions, electrons, radicals, activated molecules, or long-lasting their recombinants (e.g. O3 and H2O2) [16-18]”
3.Line 94 the author chose 40℃ and 4 h as the center point in the CCD. The Basis for the experimental design should be decided such as the batch test results.
Response: We have reinforced the materials and methods section to be more accurate about the design of experiments as shown below:
(Page 3, Line 116) “The DOE based on actual variations of independent variables and coded variables normalized to 1 or -1 is shown in Table 2 and Figure 2 to optimize the responses from the combination of two groups of design points according to multiple response algorithms in response surface methodology [27]. Since the temperature range is determined according to the set levels of a coded variable, the center point was set at 40℃ to prevent the temperature from being set to below zero or high. For CT, the center point was set at 4 hours to guarantee the effectiveness of pretreatment. All the experiments were conducted in triplicate.”
4.Line 125 The concentration of ammonium nitrogen should be analyzed in this experiment. The author mentioned that “little insights have been provided on how NTP contributes to ammonia stripping”, which indicates that NTP has potential applications in ammonia stripping. However, there is no any result about the ammonia.
Response: Our goal was to find out the optimal operating condition of NTP, which we achieved. In parallel experiments, we assessed the ammonia by capturing it from the outgoing gases. This is an extensive study on its own and we are planning to report the results separately. To avoid confusion, we revised the sentence as shown below:
(Page 2, Line 65) “Also, little insights have been provided on how NTP contributes to TN removal according to temperature and CT [24].”
5.Line 149 “pH concentration” is inappropriate.
Response: The sentence was corrected as indicated below:
(Line 167) “pH value”
6.About the effects of temperature, 20℃ is generally room temperature. 12℃ is usually below room temperature. How these temperature effects the oxidation of the pollutants. The author should give a detailed explanation.
Response: As indicated by the reviewer, it is well known that the reaction rate depends on the temperature. However, in this study, the temperature of the applied air was approximately room temperature (25 oC) while the reactor’s liquid temperature was maintained following the experimental design. It may reduce treatment efficiency slightly. For clarification, we revised our explanations precisely.
(Page 3, Line 109) “To verify the optimal operating condition of the NTP, a CCD was applied by setting the temperature (℃) of the reactor and the CT (hr) as two independent variables,”
(Page 3, Line 89) “An aeration pump (Goseong valve, YP-20A, Korea) was connected to the NTP system to supply the air of room temperature (about 25 oC) with the maximum air flowrate of 20 L/min.”
7.Line 176 the title of figure 4 was not inappropriate.
Response: As suggested by the reviewer, we revised the caption of Figure 4 as shown below:
(Page 7, Line 194) “Figure 4. Variations of TCOD and TSS according to the experimental condition of CCD. The average values of initial TCOD and TSS were 337. 5 mg/L and 171.6 mg/L, respectively.”
Reviewer #3:
- At line 164, authors state that "the same CT (4hr) at the range of temperature shows better performance since different temperatures (12 ℃, 40 ℃ and 68 ℃) recorded removal efficiency as 84.5%, 84.5%, and 20.4%, respectively" Can authors give a discussion about the 12 ℃ and 40 ℃ have the same removal efficiency?
- At line 157, authors state that "These results indicate that adequate temperature control below the central point (40 ℃, 4 hr) is helpful to obtain higher removal efficiency." Can authors explain that lower temperature have higher remove efficiency?
Response to comment 1 & 3: The reviewer’s point on context is well taken. The operational advantages of high temperature cannot be attained in this experimental setup. We fully rechecked the raw data and found no clue regarding this discrepancy. Because we employed the aeration of room air to gain economical operation, it seems to have a cooling effect for high-temperature operation and a warming effect for low-temperature operation though it is just a speculation. We are working on it to reveal the reasons for another study.
Therefore, to restrict the case to NTP and to avoid unnecessary debate, we revised the sentence as shown below. This revision may make readers focus on establishing the optimization goal rather than mechanism verification.
(Page 6, line 175) “These results indicate that adequate temperature control below the central point (40 ℃, 4 hr) is helpful for NTP to obtain better removal efficiency.”
- At line 173, authors state that "These results indicate that proper adjustment of the NTP’s operating temperature and CT at around 40 ℃ and longer than 4 hr may lead to better TCOD and TSS removal." Authors should give a temperature range and time according to the results.
Response: Indicated sentence is mentioning the effects of temperature and CT only on TCOD and TSS. Section 3.2.4 overlay plot for process optimization is actually showing the optimal range of temperature and CT as 55.2 ~ 58.2 ℃ of temperature and 1 hr 43 min ~ 2 hr 43 min of CT considering TCOD, TSS, and TN simultaneously.
4.At line 279, authors state that "T-N removal was effective at high temperatures (> 40 ℃) due to the stripping mechanism." Can authors explain the "stripping mechanism"?
Response: We added all the possible mechanisms in line 204~211 by citing available references. As we responded fully about to comment 4 from reviewer 2, we are conducting another study to verify which mechanisms are actually happening. Therefore, to prevent confusion, we revised the text as below:
(Page 11, Line 303) “TN removal was effective at high temperatures (> 40 ℃).”

Reviewer 2 Report
General comments:
This study optimized the non-thermal plasma condition on pollutant oxidation. The effects of temperature and contact time were examined by using the response surface method. The experimental design is appropriate and the results is valuable. However, there were still some mistakes in the manuscript and the related mechanism (e.g. the analysis of the radical) is missed. The author should give related discussion in detail when considering resubmission of their work.
Specific comments:
- The concept of non-thermal plasma should be explained in the Introduction section.
- Line 86 Figure 1. “Reactive chemicals” was noted in figure 1, but there was no explanation about which kind of chemicals was added.
- Line 94 the author chose 40℃ and 4 h as the center point in the CCD. The Basis for the experimental design should be decided such as the batch test results.
- Line 125 The concentration of ammonium nitrogen should be analyzed in this experiment. The author mentioned that “little insights have been provided on how NTP contributes to ammonia stripping”, which indicates that NTP has potential applications in ammonia stripping. However, there is no any result about the ammonia.
- Line 149 “pH concentration” is inappropriate.
- About the effects of temperature, 20℃ is generally room temperature. 12℃ is usually below room temperature. How these temperature effects the oxidation of the pollutants. The author should give a detailed explanation.
- Line 176 the title of figure 4 was not inappropriate.
Author Response

(The authors gave the same response as above.)

Reviewer 3 Report
- At line 164, authors state that "the same CT (4hr) at the range of temperature shows better performance since different temperatures (12 ℃, 40 ℃ and 68 ℃) recorded removal efficiency as 84.5%, 84.5%, and 20.4%, respectively" Can authors give a discussion about the 12 ℃ and 40 ℃ have the same removal efficiency?
- At line 173, authors state that "These results indicate that proper adjustment of the NTP’s operating temperature and CT at around 40 ℃ and longer than 4 hr may lead to better TCOD and TSS removal." Authors should give a temperature range and time according to the results.
- At line 157, authors state that "These results indicate that adequate temperature control below the central point (40 ℃, 4 hr) is helpful to obtain higher removal efficiency." Can authors explain that lower temperature have higher remove efficiency?
- At line 279, authors state that "T-N removal was effective at high temperatures (> 40 ℃) due to the stripping mechanism." Can authors explain the "stripping mechanism"?
Author Response

(The authors gave the same response as above.)

Round 2
Reviewer 1 Report
The authors have improved the quality of the manuscript accordingly. This manuscript may be accepted for publication.
Reviewer 2 Report
Properly revised.